# Neonatal Hypoxic-Ischemic Brain Injury Alters Brain Acylcarnitine Levels in a Mouse Model

**DOI:** 10.3390/metabo12050467

**Published:** 2022-05-22

**Authors:** Amanda M. Dave, Thiago C. Genaro-Mattos, Zeljka Korade, Eric S. Peeples

**Affiliations:** 1Department of Pediatrics, University of Nebraska Medical Center, Omaha, NE 68198, USA; amanda.dave@unmc.edu (A.M.D.); zeljka.korade@unmc.edu (Z.K.); 2Children’s Hospital & Medical Center, Omaha, NE 68114, USA; 3Child Health Research Institute, Omaha, NE 68198, USA; thiago.mattos@unmc.edu; 4Munroe-Meyer Institute for Genetics and Rehabilitation, University of Nebraska Medical Center, Omaha, NE 68198, USA; 5Department of Biochemistry and Molecular Biology, University of Nebraska Medical Center, Omaha, NE 68198, USA

**Keywords:** carnitine, encephalopathy, plasma, asphyxia, biomarker

## Abstract

Hypoxic-ischemic brain injury (HIBI) leads to depletion of ATP, mitochondrial dysfunction, and enhanced oxidant formation. Measurement of acylcarnitines may provide insight into mitochondrial dysfunction. Plasma acylcarnitine levels are altered in neonates after an HIBI, but individual acylcarnitine levels in the brain have not been evaluated. Additionally, it is unknown if plasma acylcarnitines reflect brain acylcarnitine changes. In this study, postnatal day 9 CD1 mouse pups were randomized to HIBI induced by carotid artery ligation, followed by 30 min at 8% oxygen, or to sham surgery and normoxia, with subgroups for tissue collection at 30 min, 24 h, or 72 h after injury (12 animals/group). Plasma, liver, muscle, and brain (dissected into the cortex, cerebellum, and striatum/thalamus) tissues were collected for acylcarnitine analysis by LC-MS. At 30 min after HIBI, acylcarnitine levels were significantly increased, but the differences resolved by 24 h. Palmitoylcarnitine was increased in the cortex, muscle, and plasma, and stearoylcarnitine in the cortex, striatum/thalamus, and cerebellum. Other acylcarnitines were elevated only in the muscle and plasma. In conclusion, although plasma acylcarnitine results in this study mimic those seen previously in humans, our data suggest that the plasma acylcarnitine profile was more reflective of muscle changes than brain changes. Acylcarnitine metabolism may be a target for therapeutic intervention after neonatal HIBI, though the lack of change after 30 min suggests a limited therapeutic window.

## 1. Introduction

### 1.1. Hypoxic-Ischemic Brain Injury

Neonatal hypoxic-ischemic encephalopathy (HIE), which is the clinical phenotype resulting from hypoxic-ischemic brain injury (HIBI), is one of the most serious perinatal injuries in term infants, annually affecting 1 to 8 per 1000 children in developed countries and up to 26 per 1000 in underdeveloped countries [1]. The current mainstay of therapy for affected infants is hypothermia, but despite hypothermia, many infants with HIBI still die or suffer significant neurodevelopmental impairment [2]. The future development of supplemental therapies to improve outcomes for this population will rely on identifying novel targets for therapeutic intervention. A primary feature of HIBI pathophysiology is cellular metabolic failure [3]. As such, targeting metabolic pathways such as fatty acid transport—which relies on L-carnitine and its acylcarnitine derivatives—may prove advantageous in the care of infants affected by HIBI.

### 1.2. Acylcarnitines

Fatty acids must be conjugated to L-carnitine to enter the mitochondria, and acylcarnitines function to transfer acyl groups from the cytoplasm to the mitochondria during fatty acid oxidation [4,5].The first 72 h after HIBI (including both the primary and secondary phases of brain injury) is primarily characterized by mitochondrial energy failure and the failure of the oxidative metabolism [6]. Specifically, alterations in the mitochondrial membrane and reactive oxidative species accumulation lead to changes in membrane permeability and the decoupling of oxidative phosphorylation [7]. Given the key role that the mitochondrial metabolism plays in HIBI, acylcarnitines are likely candidates for either diagnostic biomarkers or therapeutic targets.

In addition to their function in metabolic activity, acylcarnitines may play a role in neuronal membrane modification and stabilization [4]. As such, carnitine supplementation has been investigated as a possible therapeutic intervention in animal models of neonatal HIBI. An initial study demonstrated that L-carnitine administration immediately prior to HIBI was effective at reducing brain injury after HIBI in newborn rats; however, that study showed no improvement in outcomes with a single dose of L-carnitine given at 1 or 4 h after HIBI [7]. These results contrast those of a more recent study, which demonstrated moderate improvement in behavioral testing in P7 rats treated with multiple doses of an acetylated version of L-carnitine at 0, 4, 24, and 48 h after HIBI [8]. In order to better understand the optimal timing and structure of a potential intervention targeting the carnitine pathway, it will be important to understand the timing of potential changes in acylcarnitine expression following neonatal HIBI.

### 1.3. Role of Acylcarnitines in Hypoxic-Ischemic Brain Injury

A few studies have provided a profile of circulating acylcarnitines after neonatal asphyxia and HIE. In cord blood collected immediately after injury, long-chain acylcarnitines were increased in both infants that experienced asphyxia (defined by signs of hypoxia-ischemia but with a normal neurological exam) and infants diagnosed with HIE [9]. Another study did not specifically separate infants with HIE, but also demonstrated increased concentrations of C16:0, C18:0, and C18:1 acylcarnitine with decreasing umbilical arterial pH and 5 min Apgar score [10]. Lastly, in a study that used dried blood spots obtained at 3 days of life, subjects with HIE had significantly lower serum levels of long-chain acylcarnitines compared with control subjects [11], suggesting a shift in the profile from the cord blood studies. Levels of individual acylcarnitines in the brain, however, have not yet been evaluated in a neonatal model of HIBI.

We hypothesized that there would be temporal and brain-region-specific alterations in acylcarnitines after neonatal HIBI. Given that neonatal HIBI is a triphasic injury, identification of the timing of the alterations in acylcarnitines may prove advantageous in the choice and the timing of supplementation. The goal of this study was to profile regional and temporal differences in acylcarnitines after HIBI to (1) determine if changes in one or more acylcarnitine may act as a brain-specific biomarker to diagnose HIBI and (2) evaluate whether acylcarnitine changes exist in the brain after neonatal HIBI, which may suggest a metabolic target for developing future therapeutics.

## 2. Results

### 2.1. Individual Brain Acylcarnitine Changes: HIBI vs. Control

In brain regions collected 30 min after HIBI, significant increases were primarily seen in palmitoylcarnitine (C16:0) and stearoylcarnitine (C18:0) (Figure 1) compared with the controls. When restricted to only severely injured brains, the differences in C18:0 remained in the cortex (mean difference HIBI vs. control = 10.3, 95% CI 7.7–12.9, *p* < 0.0001) and striatum/thalamus (mean difference HIBI vs. control = 5.0, 95% CI 3.3–6.7, *p* < 0.0001), but the differences in C16:0 were no longer significant. C18:1 was also significantly elevated in the plasma, and while it did not reach statistical significance, there was a trend toward higher C18:1 in all three brain regions.

Expression in the HIBI group normalized in the brain by 24 h after injury, including in those with severe injury, although some differences were noted in the cerebellum at 72 h after injury (Appendix A). The plasma results were unchanged when restricted only to those with severe brain injury.

### 2.2. Total Brain Acylcarnitine Changes: HIBI vs. Control

Total acylcarnitines were also significantly higher at 30 min after injury in the plasma, but the brain regions and other time points demonstrated no differences in total acylcarnitine content (Figure 2).

### 2.3. Systemic Acylcarnitine Changes: HIBI vs. Hypoxia vs. Control

Figure 3 shows the total acylcarnitine content in all tissues and plasma of the four most altered acylcarnitines at 30 min after injury. Concentrations were higher in the HIBI group than in the controls in the brain, plasma, and muscle, while the concentration in the liver was generally lower in the HIBI group. Of note, the overall concentrations of acylcarnitines (specifically C14:0, C16:0, and C18:1) were more increased in the muscle than in any of the brain regions.

Appendix A demonstrates the acylcarnitine changes in each tissue, including the hypoxia-only groups, to aid in differentiating the effects of ischemia versus hypoxia. The HIBI-exposed animals had significantly higher C18:0 in the cortex and striatum/thalamus compared to hypoxia alone. In the plasma, significant increases were seen in HIBI versus hypoxia alone in C14:0, C16:0, and C18:1, and similar changes were seen in the muscle in C14:0 and C16:0.

## 3. Discussion

In this study, we were the first to assess specific acylcarnitine concentrations in the brain after neonatal HIBI. We found that acylcarnitine levels—specifically the expressions of palmitoylcarnitine (C16:0) and stearoylcarnitine (C18:0)—are altered in the brain 30 min after injury, but resolve by 24 h. Plasma acylcarnitine profiles were inconsistent with those demonstrated in the brain, with much of the plasma profile being driven by high concentrations in the muscle. This suggests that plasma acylcarnitines may have limited specificity to detect brain injury after hypoxia-ischemia. There is the potential to target acylcarnitine changes as an early intervention in neonatal HIBI, though the therapeutic window may be fairly narrow.

### 3.1. Acylcarnitine Function

Hypoxia-ischemia alters mitochondrial dynamics and inhibits oxidative phosphorylation and beta-oxidation [12]. Hypoxic-ischemic injury leads to the depletion of ATP and oxidative stress [13]. In addition to the generation of ATP, the mitochondria function to shuttle carnitines [14]. Hypoxia inhibits oxidative phosphorylation and beta-oxidation, which may result in accumulation of fatty acids and acylcarnitines in the mitochondria (Figure 4) and is the most likely mechanism for the elevated brain acylcarnitine levels demonstrated after HIBI in our current study.

We found that C16:0 acylcarnitine was increased in the cortex and C18:0 acylcarnitine was elevated in all three brain regions at 30 min after HIBI. In addition to its role in mitochondrial fatty acid transport, C16:0 is associated with cardiac ischemia [15]. Elevated C16:0 aggravates cellular injury after cardiac ischemia, resulting from increased reactive oxygen species production and effects on Na/K-ATPase, resulting in sodium influx into the cell, both of which are also known mechanisms of injury in neonatal HIBI [16,17]. Additionally, C16:0 has been shown to activate caspases, resulting in increased cellular apoptosis [18]. Similarly, the other long-chain acylcarnitine species found to be elevated in our model, C18:0, was also shown to alter intracellular calcium concentrations after ischemia [19]. Although both C16:0 and C18:0 were elevated after HIBI compared with controls, C18:0 was also significantly elevated compared with hypoxia only, suggesting its elevation may be specific to ischemia. Both C16:0 and C18:0 were also elevated in cord blood from neonates with asphyxia with and without HIE compared with controls [9,20].

Previous studies have assessed the potential therapeutic benefit of L-carnitine supplementation and have demonstrated benefits when given as a pretreatment or as post-treatment starting immediately after injury [8,21,22]. Although a multidose posttreatment regimen of acetyl-L-carnitine starting immediately after injury was beneficial, single doses of L-carnitine administered either at one or four hours after injury did not provide benefit [7]. Due to the multiple differences in methodology, however, it is unclear if these differences in response are due to the different chemical properties or different dose timings and frequency. Although our data do not demonstrate exactly when acylcarnitine concentrations return to baseline, the elevations were seen at 30 min but not at 24 h after injury, suggesting a potentially limited therapeutic window for acylcarnitine therapeutic targeting, which may be why the study starting therapy immediately after injury demonstrated benefit while the ones at one and four hours after injury did not. Future studies should assess brain acylcarnitine levels at multiple time points in the first 6 h after injury to better identify the precise window for potential therapeutic targeting.

### 3.2. Plasma Acylcarnitine Changes

In addition to their therapeutic potential, plasma acylcarnitine concentrations have been evaluated as potential biomarkers for HIBI. Similar to our findings, human cord blood studies have also demonstrated alterations in long-chain acylcarnitines such as C16:0, C18:0, and C18:1 after asphyxia, perinatal acidosis, and HIE [9,10]. At 72 h after injury, dried blood spots of infants with HIE had decreased long-chain acylcarnitine plasma levels compared with those of controls, which was also seen, to a lesser extent, in our animal model at 72 h [11]. Although the C16:0 changes were consistent between brain tissue and plasma, the plasma also had significant elevations of C14:0 and C18:1, which were not seen in the brain tissue. Not only were the profiles different between the plasma and brain tissue, there were also no differences appreciated in the plasma levels between those with and without severe brain injury. Similar to our findings, in normal developing rats, brain acylcarnitine profiles did not correlate with plasma acylcarnitine expression [23]. Given the discrepancy between plasma and brain acylcarnitine profiles, we then examined systemic tissues such as liver and muscle to evaluate the underlying etiology of the discrepancy.

### 3.3. Systemic Acylcarnitine Changes

We demonstrated significant elevations in C14:0 and C18:1 levels in the muscle, which was consistent with the plasma findings. Additionally, the magnitude of change in the muscle was much larger than that in the brain. The animal model used in this study resulted in systemic hypoxia but not systemic ischemia, so we also used a hypoxia-only exposure to assess for the cause of the muscle acylcarnitine changes. Hypoxia only had a limited effect on muscle tissue acylcarnitines. Because there was also a difference in anesthesia exposure between the HIBI and hypoxia-only models (HIBI was exposed to isoflurane but hypoxia only was not), this could have also led to the increases in acylcarnitine levels in the muscle tissue. Previous studies, however, showed a decrease in systemic acylcarnitines after isoflurane exposure rather than the increase seen in our study [24]. One unmeasured variable that could have led to the muscle tissue changes in the HIBI model was seizure activity, as increased muscle activity leads to acute elevations in long-chain acylcarnitines [25]. From the magnitude and pattern of acylcarnitine elevations in the muscle, it is likely that the muscle changes drove the plasma changes more than the brain. The liver acylcarnitines levels were decreased in the liver after HIBI. Although the exact cause of the decreased concentrations was outside of the scope of this study, it may have been due to the isoflurane exposure mentioned above.

These data present a preliminary investigation into the temporal acylcarnitine changes occurring after neonatal HIBI. Further assessments between 30 min and 24 h after injury are needed to address whether the acylcarnitine alterations remain elevated long enough to be a feasible target for therapy. Additionally, because this study focused on biomarker/therapeutic target discovery, future studies are needed to investigate the mechanisms underlying the acylcarnitine changes seen here. Hypoxic-ischemic injury in the neonate is also a systemic process rather than the isolated brain process that is generated in the animal model. Although the plasma findings in our study closely mimic those seen in humans, there are certainly some limitations to translating our findings in the animal model to humans. Lastly, we did not monitor for seizure activity, and therefore could not fully evaluate the potential causes for muscle acylcarnitine changes in this study.

## 4. Materials and Methods

### 4.1. IACUC Statement

This study was approved by the Institutional Animal Care and Use Committee at the University of Nebraska Medical Center. The guidelines for proper animal use and care were strictly followed according to the Guide for the Care and Use of Laboratory Animals of the National Institute of Health.

### 4.2. Animal Model of Hypoxia-Ischemia

Postnatal day 9 CD1 mouse pups of both sexes were anesthetized with isoflurane and randomized to HIBI induced by carotid artery ligation followed by 30 min at 8% oxygen, or sham surgery followed by 30 min at 21% oxygen. Pups were maintained at normothermia throughout surgery and recovery. Each group was further divided into three sub-groups for tissue collection at 30 min, 24 h, or 72 h after injury (12 animals/group at each time point). Whole blood was obtained and centrifuged to extract plasma, and brain tissue was immediately extracted and dissected prior to storage at −80 °C.

Some regions of the brain are more sensitive to HIBI than others, including the cortical white matter, basal ganglia, hippocampus, and thalamus [26,27]. In the rodent carotid artery ligation HIBI model—and to a lesser extent in human infants with HIBI—the cerebellum and other structures supplied by the posterior circulation tend to be spared [3]. As such, brain tissue was dissected into three regions for separate analyses: cortex, cerebellum, and striatum/thalamus.

Because organs other than the brain also likely significantly contribute to plasma acylcarnitine concentrations, liver and skeletal muscle were also collected from the HIBI and control groups. A third exposure was also added, which included animals not exposed to anesthesia or surgery but received 30 min of 8% oxygen, in order to understand the effects of hypoxia alone versus hypoxia-ischemia.

### 4.3. Tissue Preservation and Analysis

The tissue was placed in phosphate-buffered saline (PBS) with antioxidants (1% butylated hydroxytoluene/triphenylphosphine) and homogenized. Aliquots of sample were taken for protein quantification by BCA, interleukin-6 (IL-6) quantification, and acylcarnitine profiling. Due to concerns regarding the variable levels of injury in this model, we evaluated the severity of injury in the ipsilateral cortex by IL-6 to ensure that there was significant and consistent injury in the HIBI group. IL-6 was measured by DuoSet ELISA (R&D Systems, Minneapolis, MN, USA) per the manufacturer’s instructions. The cutoff for severe injury was chosen as the 75th percentile for IL-6 expression at each time point.

### 4.4. Mass Spectrometry

The acylcarnitine profile was assessed by LC-MS/MS as described previously [28,29]. Briefly, samples were spiked with a known amount of d_3_-palmitoylcarnitine as the internal standard, and the lipids were extracted. After extraction, the lysates were injected onto a column (Phenomenex Luna Omega C18, 1.6 μm, 100 Å, 2.1 mm × 100 mm) using acetonitrile/water (solvent B: 90:10 *v*/*v*, 0.1% *v*/*v* formic acid, and 10 mM ammonium formate) and acetonitrile/water (solvent A: 10:90 *v*/*v*, 0.1% *v*/*v* formic acid and 10 mM ammonium formate) as mobile phases. The total run time was 5.5 min at a flow rate of 500 μL/min. The LC gradient was set to: 40% B for 0.5 min; 40% to 90% B for 1.5 min; 90% B for 2 min; 90% to 100% B for 0.1 min; 100% B for 0.8 min; 100% to 40% B for 0.1 min; 40% B for 0.7 min. Individual acylcarnitines were analyzed by an Acquity UPLC system equipped with an ANSI-compliant well-plate holder coupled to a Thermo Scientific TSQ Quantis mass spectrometer, equipped with an ESI source. Analyses were performed in positive ion mode and individual acylcarnitine species were measured by selective reaction monitoring (SRM) using transitions of the precursor ion (as [M+H]^+^) to the respective product ions with 85 *m*/*z*.

The acylcarnitine levels in each brain region, plasma, and systemic tissues were compared between HIBI, hypoxia only, and control animals at each of the three time points, separately. We compared the three groups using two-way ANOVA, followed by Tukey’s test, for multiple comparisons. A *p*-value < 0.05 was considered significant.

## 5. Conclusions

This study demonstrated that acylcarnitine levels are altered in the brain, plasma, and muscle tissue at 30 min after HIBI. Although our plasma results mimic those seen in humans, our data suggest that the plasma acylcarnitine levels do not reflect brain injury, but rather more likely reflect systemic acylcarnitine changes. Better understanding of aberrations in brain metabolism and the role of acylcarnitines after HIBI may provide an opportunity to identify targets for potential interventions, though the lack of change after 30 min in this study suggests that there may be a limited therapeutic window. Further studies will be required to delineate biochemical interventions, including isolation of mitochondria, to further understand beta oxidation and the potential of acylcarnitines for neuroprotection.

## Figures and Tables

**Figure 1 metabolites-12-00467-f001:**
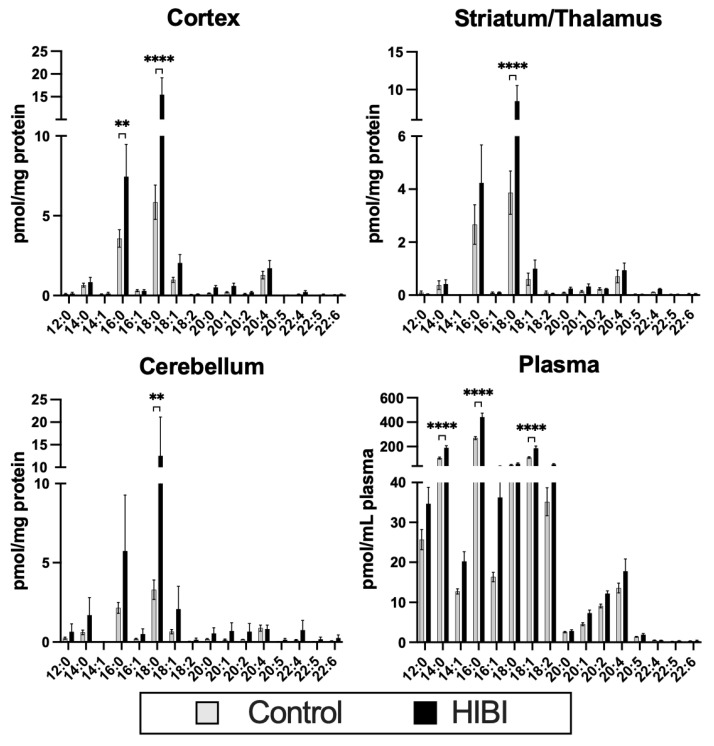
Individual acylcarnitine levels in brain regions and plasma at 30 min after HIBI compared with sham surgery controls. ** *p* < 0.01, **** *p* < 0.0001.

**Figure 2 metabolites-12-00467-f002:**
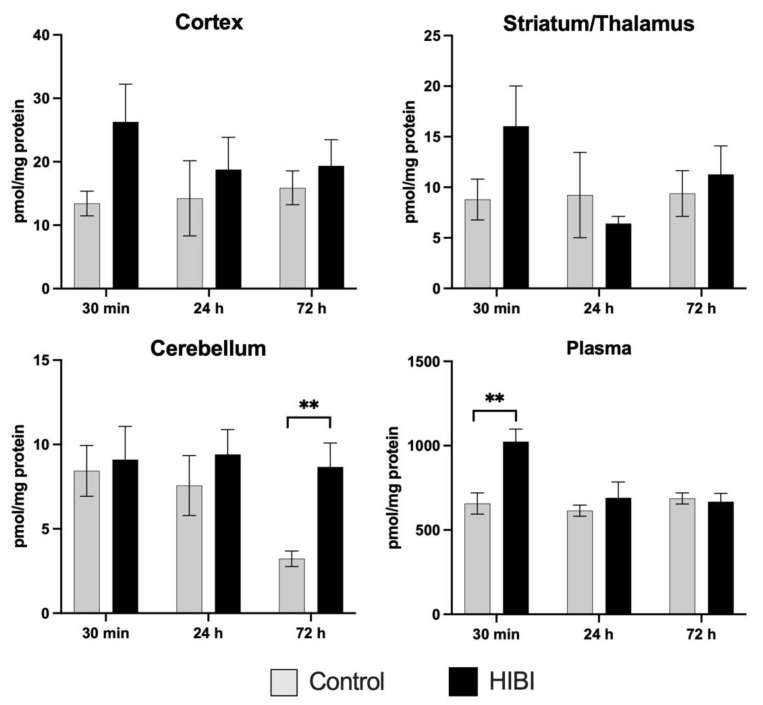
Total acylcarnitine levels for brain regions and plasma over time. ** *p* < 0.01.

**Figure 3 metabolites-12-00467-f003:**
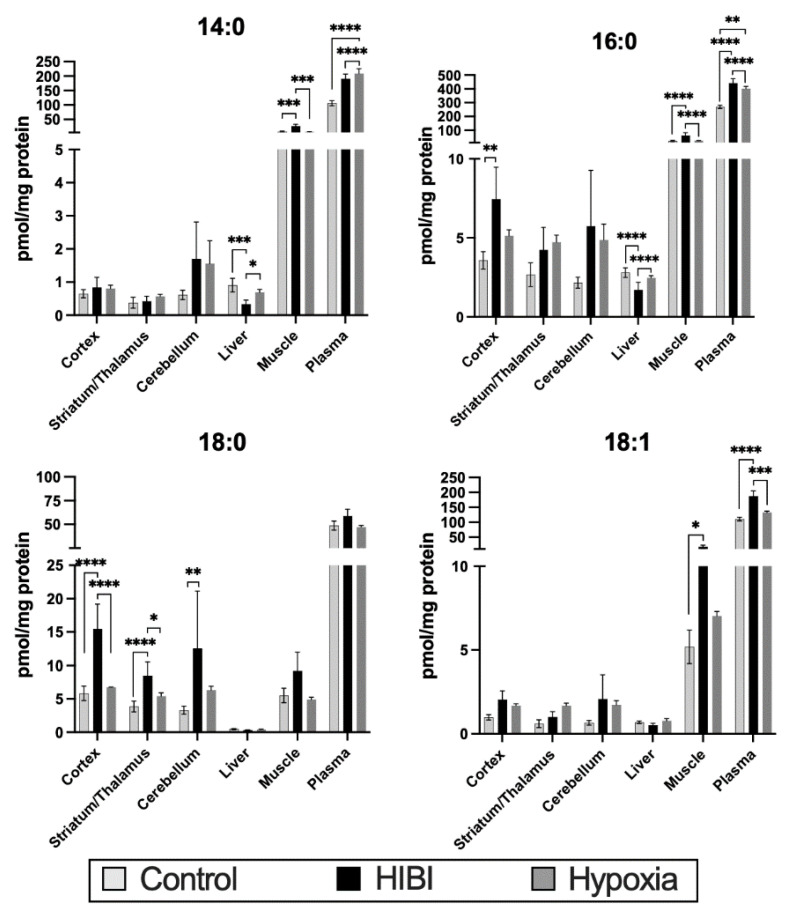
Levels of the four primarily altered acylcarnitines in the different tissues analyzed at 30 min after HIBI compared with hypoxia or controls. * *p* < 0.05, ** *p* < 0.01, *** *p* < 0.001, **** *p* < 0.0001.

**Figure 4 metabolites-12-00467-f004:**
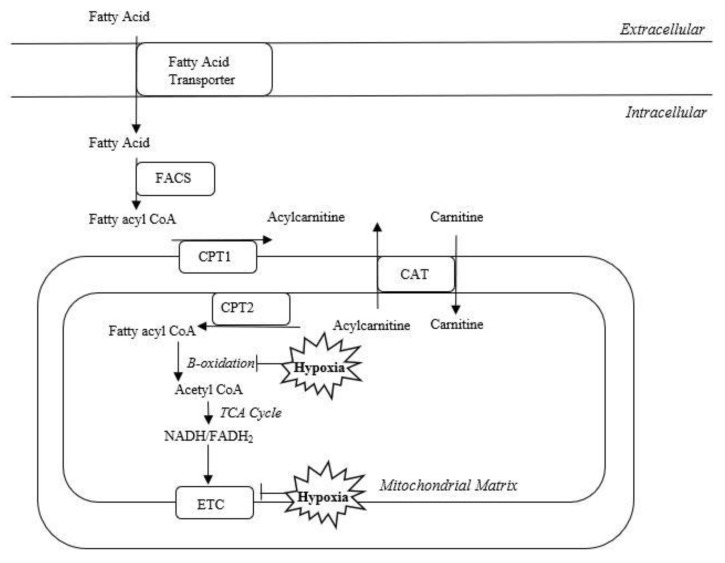
Cell hypoxia results in the inhibition of oxidative phosphorylation and beta-oxidation in the mitochondria, leading to the accumulation of fatty acids and acylcarnitines. Fatty-acyl-CoA synthase (FACS); carnitine palmitoyltransferase 1 (CPT1); carnitine palmitoyltransferase 2 (CPT2), carnitine acetyltransferase (CAT); electron transport chain (ETC).

## Data Availability

Data are contained within the article or Appendix A.

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
