# Peer review of "Neonatal Hypoxic-Ischemic Brain Injury Alters Brain Acylcarnitine Levels in a Mouse Model"

_metabolites, 2022, doi:10.3390/metabo12050467_

Round 1

Reviewer 1 Report

This study investigated the temporal acylcarnitine changes occurring after neonatal hypoxic-ischemic brain injury in mice. The authors need to address several issues to enhance their work.

The manuscript is prepared incomprehensibly, making it challenging to understand the content. Please clearly state the purpose of this study. Moreover, my impression is that the authors show a little too few results to draw conclusions.

Materials and methods should be divided into subsections.

Figures are pasted into the text without captions.

Please rewrite the whole manuscript, divide it into subchapters, add graphics showing the studied metabolic pathways affected by hypoxic-ischemic brain injury, table with types of acylcarnitines; add figure captions with description (description of x and y-axis, n number, statistical analysis).

Reviewer 2 Report

The research article entitled “Neonatal Hypoxic-ischemic Brain Injury Alters Brain Acylcarnitine Levels in a Mouse Model” by Amanda M. Dave and coworkers observed acylcarnitine levels in brain, plasma, and muscle tissue during hypoxic- ischemic brain injury. The MS is very short but précised. Although, the data is preliminary but have much impact. Some minor comments may be address to improve quality of MS.

It is better to polish the abstract and highlights all the information and topics covered in the manuscript.

Introduction: Well written, however can give recent data about previous finding (only two references from last 3 years).

Mitochondria are a major target in hypoxic/ischemic injury; Authors can correlate and discussed these finding in relation with mitochondrial dysfunction/function.  

Round 2

Reviewer 1 Report

The authors have satisfactorily addressed most of my concerns. The only remaining issue I have with the abstract are seemingly minor: The abbreviation "AC" is not introduced and explained in the abstract.